# IER5 Promotes Ovarian Cancer Cell Proliferation and Peritoneal Dissemination

**DOI:** 10.3390/cancers17040610

**Published:** 2025-02-11

**Authors:** Jayaraman Krishnaraj, Sayaka Ueno, Moe Nakamura, Yuko Tabata, Tatsuki Yamamoto, Yoshinori Asano, Tomoaki Tanaka, Tomohisa Kuzuyama, Hideyuki Saya, Rieko Ohki

**Affiliations:** 1Laboratory of Fundamental Oncology, National Cancer Center Research Institute, Tsukiji 5-1-1, Chuo-ku, Tokyo 104-0045, Japan; kjayaram@ncc.go.jp (J.K.); mo.nakamu29@gmail.com (M.N.); yutabata@ncc.go.jp (Y.T.); tayamamo@ncc.go.jp (T.Y.); y.asano1242@gmail.com (Y.A.); 2Department of Genomic Medicine, Fujita Health University, Toyoake 470-1192, Japan; sayaka.ueno@fujita-hu.ac.jp (S.U.); hsaya@fujita-hu.ac.jp (H.S.); 3Graduate School of Agricultural and Life Sciences, The University of Tokyo, Tokyo 113-8654, Japan; utkuz@g.ecc.u-tokyo.ac.jp; 4Department of Molecular Diagnosis, Chiba University School of Medicine, Chiba 260-0856, Japan; tomoaki@restaff.chiba-u.jp; 5Graduate School of Medicine, Keio University, Tokyo 160-0016, Japan

**Keywords:** ovarian cancer, IER5, HSF1, HSP, metastasis

## Abstract

Epithelial ovarian cancer (OC), the most common type of OC, frequently harbors p53 mutations. In this study, we found that the immediate early response 5 gene (*IER5*), a p53 target gene, is overexpressed in OC cells. Importantly, OC cells found floating in the ascites had a higher expression of *Ier5* than the parent strain, suggesting a role for IER5 in metastasis. Our previous research demonstrated that IER5 activates heat shock factor-1 (HSF1), a key transcription factor that regulates heat shock protein (*HSP*) genes by promoting the formation of a hypo-phosphorylated active form of HSF1. This activation of HSF1 by IER5 leads to the transcriptional activation of *HSP* genes, which help protect cells from stress. Here we showed that knockdown of *Ier5* reduced the proliferation of OC cells as well as the induction of HSPs. These results indicate that the IER5-HSF1-HSP pathway contributes to the proliferation and peritoneal dissemination of OC cells.

## 1. Introduction

Ovarian cancer (OC) is one of the most lethal gynecological cancers [1]; 90% of the reported cases are of epithelial origin, while other forms such as those of germ-cell origin constitute the remaining [2]. GLOBOCAN 2020 reported a global incidence of 313,959 cases of OC, with a mortality rate of 66% in 2020 [3]. Treatment for OC involves the combination of surgical removal of the tumor mass followed by the application of chemotherapeutic drugs, usually a combination of two or more cisplatin-based drugs [4]. The overall 5-year relative survival rate is only 21% for distant stages, while the survival rate for local stages is up to 92% [3,5]. Some of the main reasons for this high mortality rate include the lack of early diagnosis and a high rate of relapse, mainly due to drug resistance [6]. In recent years, precision medicine involving patient-derived xenografts or organoid models and drug screening has become a very advanced method of treating these cancers [4,6].

Heat shock factor 1 (HSF1) is a transcription factor that functions as a master regulator of the proteotoxic stress and heat shock responses to protect cellular proteins [7]. HSF1 has been recently reported to be involved in the tumorigenesis and progression of various cancers such as ovarian, pancreatic, breast, and hepatic cancers [8,9,10]. During cancer treatment, HSF1 can reprogram the proteome to protect cancer cells from stress, which can lead to multi-drug resistance [11]. HSF1 has also been reported to induce the epithelial–mesenchymal transition in a spheroid model of ovarian cancer following transforming growth factor-β treatment [12]. Similarly, HSF1 has been shown to be required for cancer cell growth in both in vitro and in vivo models of ovarian cancer [13].

Previously, we reported that immediate early gene 5 (IER5), a transcriptional target of tumor suppressor p53, promotes tumorigenesis and confers resistance against internal and external stresses via activation of HSF1 [14]. Activation of HSF1 generally involves its hyper-phosphorylation on certain Ser/Thr residues by protein kinase A, casein kinase 2, polo-like kinase 1, mitogen-activated protein kinase, and others [7]. However, we previously reported a novel mechanism of HSF1 activation involving IER5, which facilitates the generation of a hypo-phosphorylated, activated form of HSF1 that can contribute to tumorigenesis. This mechanism involves the formation of a ternary complex of IER5 with protein phosphatase 2A (PP2A) and HSF1. PP2A dephosphorylates HSF1 in this complex at key serine and threonine residues, including Ser121, Ser307, Ser314, Thr323, and Thr367, which normally keep HSF1 inactive. Thus, this hypo-phosphorylation form of HSF1 is activated [14,15]. By this mechanism, IER5 induces the expression of heat shock proteins (HSPs), which in turn protect cells from proteomic stress [7,14].

High *IER5* expression is associated with poor prognosis in bladder, breast, brain, and glioma patients [14,16]. We also found that *IER5* is overexpressed in various cancers, the highest of which was in ovarian cancer [14]. These findings suggest that IER5 may be involved in the development and progression of ovarian cancer. Since IER5 also contributes to the survival and proliferation of cancer cells in suspension [14], we wanted to examine whether IER5 is also involved in the metastasis of ovarian cancer cells. In this study, we created a mouse model of ovarian cancer peritoneal metastasis to analyze the potential role of IER5. We showed that both IER5 and HSF1 are essential for ovarian cancer cell proliferation in both adherent and suspension conditions. We also found that the expression of various HSF1 target genes is dependent on both IER5 and HSF1 in ovarian cancer cells. These results revealed that IER5 induces HSP expression via HSF1 and plays an important role in ovarian cell carcinogenesis and proliferation.

## 2. Materials and Methods

### 2.1. Cell Culture

ID8 cells were obtained from K. F. Roby (University of Kansas Medical Center) and cultured in DMEM medium supplemented with 4% FBS, insulin (5 μg/mL), transferrin (5 μg/mL), and sodium selenite (5 ng/mL) (ITS; Sigma-Aldrich, St. Louis, MO, USA). p53-def-MOSE and T-Ag-MOSE cells were obtained from the JCRB Cell Bank (Ibaraki, Japan) and were cultured in DMEM supplemented with 10% FBS. HM-1 cells were obtained from RIKEN BRC and were cultured in RPMI supplemented with 10% FBS. Mouse ovarian tumor-initiating MOV cells were obtained from T. Motohara (Keio University) and were cultured in DMEM-F12 supplemented with 10% FBS. All cells were maintained in 5% CO_2_ at 37 °C.

### 2.2. Viral Vectors

The plasmid pMXs-IRES-GFP (kindly provided by T. Kitamura, The University of Tokyo) was used to generate a GFP-expressing retrovirus, which was transduced into ID8 cells. Infected cells were then sorted using a MoFlo XDP cell sorter (Beckman Coulter, Miami, FL, USA) to collect GFP-positive (ID8G) cells.

### 2.3. Generation of ID8-Om2 Cells

ID8G cells (5 × 10^6^) suspended in 300 µL of PBS were injected i.p. into 7- to 9-week-old female C57BL/6J mice. Between 90 and 120 days after cell injection, the omentum was isolated and dissociated by incubation for 1 h at 37 °C with collagenase (300 U/mL). Red blood cells in the tissue digest were lysed by exposure to ammonium chloride. The single-cell suspension thus obtained was sorted via a MoFlo XDP cell sorter to isolate CD45^-^GFP^+^ cells, which were cultured for several days. The resulting ID8-Om1 cells were then similarly injected i.p. into recipient mice. Subsequently, CD45^-^GFP^+^ cells were isolated from the omentum and designated ID8-Om2 cells [17].

### 2.4. In Vivo Expression of Ier5

ID8G cells (5 × 10^6^) were suspended in 300 µL of PBS and then injected i.p. into 7- to 9-week-old female C57BL/6J mice. When ascites had been sufficiently formed, it was collected and CD45^-^GFP^+^ cells were isolated via a MoFlo XDP cell sorter. The collected cells were used for analysis of *Ier5* expression.

### 2.5. RNA Extraction and RT-qPCR Analysis

Total RNA was extracted from cells by TRIzol reagent (Invitrogen, Carlsbad, CA, USA) and subjected to reverse transcription with ReverTra Ace qPCR RT Master Mix (Toyobo, Osaka, Japan). The resulting cDNA was analyzed by qPCR using TB Green Premix Ex Taq II (Takara Bio, Shiga, Japan) for the SYBR Green system or PrimeTime Gene Expression Master Mix (Integrated DNA Technologies, Coralville, IA, USA) for the TaqMan system. For *Hspa1a* (Mm PT.58. 33332390.g), *Dnajb1* (Mm PT.58.23437403) and *Hspb2* (Mm PT.58. 13396417.g), custom-designed TaqMan probes were purchased from Integrated Technologies. For *Ier5* and *actin*, custom-designed TaqMan probes from Sigma-Aldrich. Transcript levels were normalized to *β-actin* mRNA. Primer sequences used for expression analysis are listed in Table 1.

### 2.6. Animal Experiments

Animal care and procedures were performed in accordance with the guidelines of Keio University. Female immunocompetent (C57BL/6J) mice at 6 to 8 weeks of age were obtained from Charles River Japan (Atsugi, Japan). Mice were killed with the use of isoflurane before becoming moribund. All animal experimental protocols were approved by the Keio University Ethics Committee for Animal Experiments.

### 2.7. Transfection

Transient transfections were performed using Lipofectamine Plus reagent (Life Technologies, Carlsbad, CA, USA). siRNAs were introduced using OptiMEM and RNAiMAX (Invitrogen, Carlsbad, CA, USA). Control (D-001810-10-50), ON-TARGETplus *Ier5*-targeting (L-043689-01 and L-043689-01-005), and *Hsf1*-targeting (L-040660-01-0010) siRNA pools (each consisting of 4 siRNAs) were purchased from Dharmacon Research (Lafayette, CO, USA). Similarly, overexpression plasmids (for IER5-Flag) were transfected using OptiMEM and Lipofectamine 2000 reagent. The same amount of empty vector DNA was added to the control plates.

### 2.8. Cell Growth Assay

MOV or HM-1 cells (1 × 10^4^ per well) were grown in 24-well adherent tissue culture plates (Corning, Tewksbury, MA, USA) and transfected with 20 nM of *Ier5*/*Hsf1*-targeting or control siRNAs (Dharmacon, Lafayette, CO, USA). Then, 48 h later, cell numbers were counted.

### 2.9. Cell Proliferation Assay

MOV or HM-1 cells (1 × 10^3^ per well) were transferred to 96-well adherent tissue culture plates (Corning) for adherent mode or ultralow attachment 96-well plates (Corning) for suspension mode, and *Ier5*-targeting or control siRNAs were introduced. As MOV cells failed to attach to Corning plates, we used advanced μCLEAR plate (655983, Greiner Bio-One, Frickenhausen, Germany) to grow MOV cells in suspension mode. Cells were collected from 4–6 wells of each group in a 0, 1, 2, and 3-day time course after transfection, and cell proliferation was assayed with the CellTiter-Glo Luminescent Cell Viability Assay Kit (Promega, Madison, WI, USA) and a multimode plate reader (EnVision; Perkin-Elmer, Boston, MA, USA) or Wallac 1420 ARVOsx plate reader (PerkinElmer, Norwalk, CT, USA).

### 2.10. *Western Blotting Analysis*

Cells were lysed in a lysis buffer containing 50 mM Tris-HCl (pH 8.0), 1% NP40, 250 mM NaCl, 1 mM DTT, 5 mM EDTA, and 1 mM protease inhibitor (PMSF, aprotinin and leupeptin). The lysates were then treated with an equal volume of 4X SDS buffer (0.4 M Tris-HCl (pH 6.8), 8% SDS, 4% (*v*/*v*) glycerol, and 0.04% bromophenol blue). Whole cell lysates were subjected to protein quantification and analyzed by Western blotting. Antibodies used in this study: anti-IER5 rabbit polyclonal antibody (HPA029894, dilution ratio 1:5000) was purchased from Merck, anti-HSF1 rabbit polyclonal antibody (ADI-SPA-901, dilution ratio 1:10,000) was from Enzo Life Sciences, anti-DDDDK rabbit antibody (PM020, dilution ratio 1:3000) was from MBL, and anti-actin (clone C4) mouse monoclonal antibody (MAB1501, dilution ratio 1:5000) was from Millipore.

### 2.11. Statistical Analysis

Data were calculated and shown as the mean ± SD. In the figures, comparisons between the samples were performed by Student’s *t* test. Statistical significance was defined as *p* < 0.05.

## 3. Results

### 3.1. IER5 Family Genes Are Amplified or Overexpressed in Ovarian Cancer and Are Related to Poorer Prognosis

Previously, we reported that ovarian cancer has the highest degree of *IER5* overexpression among various cancer types [14]. In addition to *IER5*, *IER5L* and *IER2*—two homologues of IER5 with structural similarity [18]—are also highly expressed and promote tumor progression, metastasis, and invasion in various cancers including prostate cancer, non-small cell lung cancer, melanoma, and colorectal cancer [19,20,21,22]. Previously, we reported that IER5 contributes to tumorigenesis by recruiting PP2A phosphatase to HSF1, resulting in the dephosphorylation at inhibitory phosphorylation sites and the generation of a novel hypo-phosphorylated active form of HSF1 [14]. IER2 and IER5L have also been reported to interact with PP2A and to generate hypo-phosphorylated active HSF1 [23]. *IER5* and its family members *IER5L* and *IER2* have undergone genetic alteration in ovarian cancer, including amplification and high expression, as shown in Figure 1A. It is expected that expression of the *IER5* family genes is elevated in many ovarian cancer samples. Therefore, only several are classified as having high mRNA expression based on the strict criterion of “mRNA high”, defined as greater than two standard deviations above the mean. At least one *IER5* family gene is overexpressed or amplified in 22.5% of ovarian cancers, suggesting that this family of genes is involved in ovarian cancer promotion. In addition, the overall survival of ovarian cancer patients having alterations in *IER5* family genes is significantly lower than those without alterations, indicating that these genes are involved in cancer progression (Figure 1B). We further found that expression of the HSF1 target genes *HSPA1A* and *HSPA1B* was significantly higher in ovarian cancer samples with high *IER5* family gene expression compared to samples with low *IER5* family gene expression (Figure 1C,D), suggesting that the IER5 family gene–HSF1–HSP family gene axis may play an important role in ovarian cancer.

Ovarian cancer can originate from either ovarian surface epithelial (OSE) cells or fallopian tube epithelial (FTE) cells. We therefore analyzed the expression of *Ier5* family genes derived from OSE (Figure 1E) and FTE (Figure 1F) using data from a published mouse model of ovarian cancer [24]. We found that *Ier5* expression is elevated in ovarian cancer derived from OSE but not FTE, whereas both *Ier5l* and *Ier2* were upregulated in those from FTE but not OSE. These results collectively suggest that IER5 family genes may promote the formation and progression of ovarian cancer.

### 3.2. IER5 Is Highly Expressed in Ovarian Cancer Cells

Since we found significant elevation of *Ier5* mRNA in an ovarian cancer mouse model derived from OSE, we then decided to focus on the function of *Ier5* in ovarian cancer cells derived from OSE. We first analyzed *Ier5* mRNA levels in normal mouse OSE and cancerous mouse OSE cells. We chose two immortalized normal mouse OSE cell lines: p53 MOSE cells (from *p53*-deficient mice) and T-Ag-MOSE cells (expressing SV40 T antigen) and three mouse OSE cancer cells: MOV, ID8G, and HM-1 cells (Figure 2A). MOV cells were established from epithelial cell-adhesion molecule-positive cells derived from the ovaries of C57BL6 mice. These cells were transiently depleted of p53, followed by transfection with c-Myc and K-ras^G12V^ to generate stem-like tumor-initiating cells that give rise to lethal ovarian tumors [25]. ID8 cells are one of the 10 clones established by Roby et al. that have the highest tumor-forming capacity [26]. HM-1 cells are highly metastatic cells established by Hashimoto et al. from the lung metastasis of an ovarian primary tumor in B6C3F1 mice [27]. We found significantly higher levels of *Ier5* mRNA expression in all ovarian cancer cells (MOV, ID8G, and HM-1 cells) compared to normal ovarian cells (p53 MOSE and T-Ag-MOSE cells) (Figure 2B). We also isolated ID8-Om2 cells, which are derived from the peritoneal dissemination of ID8G cells (in the omentum). We observed an almost two-fold increase in *Ier5* mRNA expression in ID8-Om2 compared to ID8G cells (Figure 2C), indicating a possible role of IER5 in peritoneal dissemination and metastasis. This notion was further confirmed by the multi-fold upregulation of *Ier5* expression in ID8G cells from ascites isolated from mice compared to ID8G cells grown in vitro (Figure 2D). These results suggest that IER5 promotes tumorigenesis and metastasis of ovarian cancer.

### 3.3. IER5 Is Important for Ovarian Cancer Growth and Is Involved in the Induction of HSPs

To further investigate the role of IER5 in ovarian carcinogenesis, we knocked down *Ier5* expression in HM-1 and MOV cells, two ovarian cancer cell lines having higher *Ier5* expression, and analyzed the impact on cell growth (Figure 2B). Compared to MOV and HM-1 cells, ID8G cells showed lesser IER5 expression (as shown in Figure 2B). Furthermore, the efficiency of *Ier5* siRNA knockdown was also too low in ID8G and ID8-Om2 cells and did not have a significant effect on cell number (Appendix A). Therefore, we did not include ID8G cells for further analysis. In MOV and HM-1 cells, knockdown was achieved using siRNA targeting *Ier5,* and cell numbers were counted 48 h later. *Ier5* knockdown significantly reduced the numbers of both HM-1 and MOV cells compared to the siCtrl-treated group (Figure 3A–D; Appendix A). We further conducted a time-course cell proliferation assay using CellTiter-Glo Kit (Promega) to assess the effect of *Ier5* knockdown on cell proliferation. Silencing of *Ier5* expression inhibited the proliferation of these cells both in adherent and suspension cultures (Figure 3E,F). These results indicate the importance of IER5 for the proliferation of ovarian cancer cells under both adherent and suspension conditions.

### 3.4. IER5 Induces Transcription of HSPs

Previously, we showed that IER5 contributes to tumorigenesis by inducing various HSPs in epithelial lung cancer (H1299) and transformed embryonic kidney cells (293T) [14]. We hypothesized that the same mechanism might be at work in ovarian cancer. To assess the impact of *Ier5* knockdown on HSF1 target genes, we analyzed the expression of HSPs in HM-1 and MOV ovarian cancer cells following knockdown. Our results showed that the expression of key HSPs such as *Hspa1a*, *Hspa1b*, and *Dnajb1* was downregulated when *Ier5* was knocked down compared to si-Control (Figure 4B,D). *Ier5* knockdown was confirmed by analysis of *Ier5* mRNA and protein expression (Figure 4A,C; Appendix A). These results were further validated by overexpression of IER5 in HM-1 cells, whereupon mRNA levels of these HSPs (*Hspa1a*, *Dnajb1,* and *Hspb2*) were increased significantly (Figure 4E,F). These results collectively indicate that IER5 induces the transcription of HSPs.

### 3.5. IER5 Activates HSF1 via Dephosphorylation and Upregulates Its Target Genes in OC Cells

IER5 is known to generate a hypo-phosphorylated active form of HSF1 in various cancer cells [14], distinct from the canonical hyper-phosphorylated active form [7]. We analyzed the effect of *Ier5* knockdown on HSF1 phosphorylation in HM-1 cells by western blotting, and observed that the HSF1 band was shifted upwards (Figure 5A). This indicates that *Ier5* knockdown abrogated HSF1 dephosphorylation. Next, to confirm whether IER5 mediates its tumorigenic effects via HSF1, we knocked down *Hsf1* in HM-1 cells and examined cell proliferation and HSP levels. HM-1 cell proliferation was quantified by CellTiter-Glo Luminescent Kit (Promega) under adherent and suspension conditions. *Hsf1* knockdown was confirmed by RT-PCR and western blotting (Figure 5C,D). *Hsf1* knockdown suppressed the proliferation of HM-1 cells when in suspension, and less efficiently when adherent (Figure 5B). Further, expression of HSPs (*Hspa1a*, *Hspa1b*, and *Dnajb1*) was decreased similar to that caused by *Ier5* knockdown (Figure 5E). These results collectively suggest that IER5 is required for HSF1 activation and induction of HSPs in OC cells.

## 4. Discussion

*IER5* and its family members *IER5L* and *IER2* have been reported to be overexpressed in many cancers including ovarian, hepatic, lung, and gastric cancers [14,18,19,20,21]. *IER5* expression is associated with poor prognosis in glioma, bladder, and breast cancer patients [14,16]. IER5L has been shown to be a prognostic marker for non-small cell lung cancer, with higher *IER5L* expression seen in patients with relapse versus no relapse [18]. Further, *IER5L* depletion has been shown to reduce the growth, migration, and invasiveness of prostate cancer cells, indicating its importance for prostate cancer progression [19]. Similarly, IER2 has also been shown to promote invasion and metastasis in melanoma and colorectal cancer [20,21]. We now have shown that amplification and higher expression of the IER5 family genes are associated with poorer prognosis of ovarian cancer patients. Furthermore, analysis of data from an ovarian cancer mouse model seeded by OSE or FTE showed that higher expression of *Ier5* is observed in ovarian cancer cells of OSE origin, whereas higher expression of *Ier5l* and *Ier2* is observed in those derived from FTE. We also observed high expression of *Ier5* gene expression in mouse ovarian cancer cells derived from OSE (MOV, ID8G, and HM-1 cells). We also reported a two-fold higher expression of *Ier5* in metastatic tumor cells (ID8-Om2 cells) compared to original tumor cells (ID8G), suggesting a role for IER5 in metastasis. Further, using siRNA knockdown we have shown that IER5 is required for the proliferation and growth of ovarian cancer (HM-1 and MOV) cells.

Previously, we and others have reported that IER5 induces the transcription of multiple HSPs in response to various stresses including heat shock [14,28,29]. We also observed downregulation of key HSPs including *Hspa1a*, *Hspa1b*, and *Dnajb1* upon *Ier5* knockdown in HM-1 and MOV ovarian cancer cells. These results are consistent with our results showing that IER5 overexpression resulted in increased expression of key HSPs in HM-1 cells. This activation of HSPs was driven by HSF1, which is activated via IER5-mediated dephosphorylation. In addition to IER5, IER2 has also been shown to interact with PP2A, which dephosphorylates and activates HSF1 [17]. Finally, we showed that this IER5-HSF1-HSP axis is vital for cancer cell growth: *Hsf1* knockdown resulted in reduced ovarian cancer cell (HM-1) proliferation and as well as decreased HSP expression, indicating that IER5 regulates HSPs via HSF1. To summarize, the IER5-HSF1-HSP axis is vital for ovarian cancer cell proliferation and metastasis.

## 5. Conclusions

Ovarian cancer, one of the most lethal gynecological cancers, is often treated with a combination of surgery and chemotherapy using platinum compounds. Recently, PARP inhibitors have also shown promising results in ovarian cancer patients. However, these results are short-lived, and the cancer typically reoccurs [30]. As HSF1 is a master transcription factor regulating genes involved in protecting cells against proteotoxic stress, it also controls various genes related to cancer cell survival. HSF1 drives a transcriptional program in cancer cells that is different from the canonical heat shock response [31]. Some of the genes upregulated by HSF1 in cancer cells protect the cancer cells against the toxic effects of the therapeutic drugs, which can lead to drug resistance [32]. Our results support the idea that the IER5–HSF1 axis promotes ovarian cancer cell survival as well as tumor progression. Further, since other IER5 family genes (IER2 and IER5L) have also been reported to regulate HSF1 activity, this family of genes may collectively be involved in the progression of ovarian cancer via the HSF1 pathway. However, further studies including animal studies are warranted to further elucidate their roles in cancer.

## Figures and Tables

**Figure 1 cancers-17-00610-f001:**
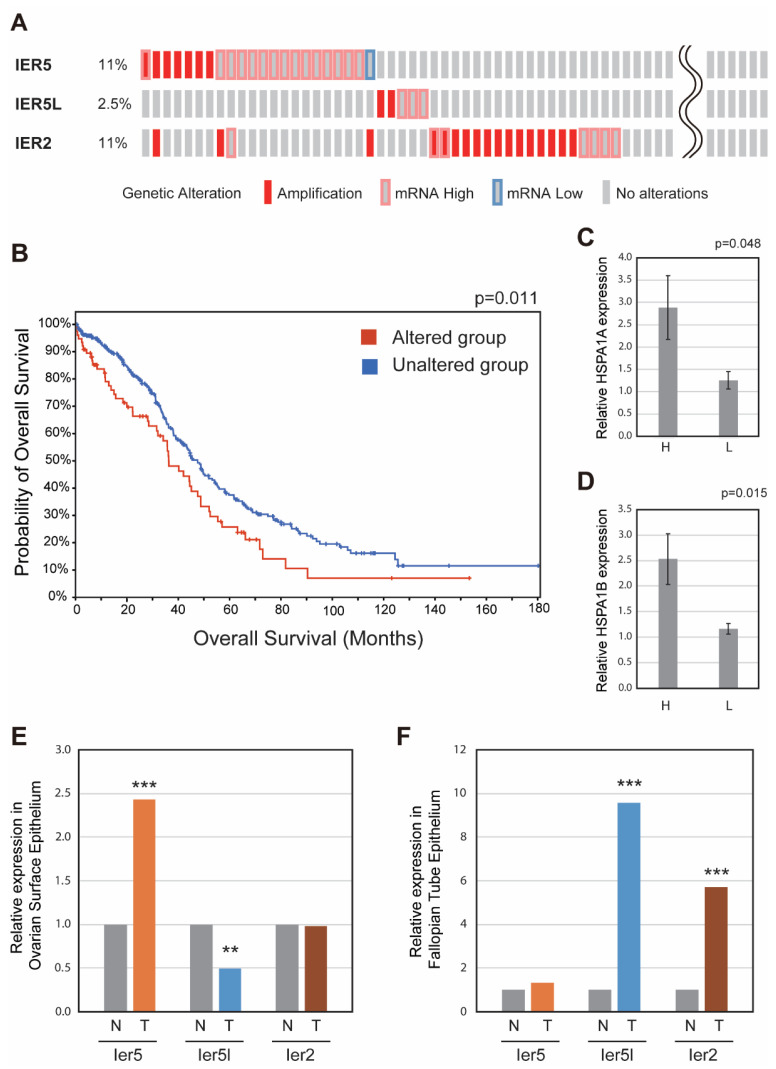
IERs are altered in ovarian cancer and impact patient survival. (**A**) Genetic alterations including gene amplification and high/low mRNA expression of *IER5*, *IER5L*, and *IER2* in human ovarian cancers (http://cbioportal.org). “Amplification” refers to high-level amplification (more than 8 copies), while “high mRNA expression” is defined as expression greater than 2 standard deviations above the mean. (**B**) Patient overall survival in the IER5-altered and unaltered groups. Logrank test was used to determine the ‘*p*’ value (http://cbioportal.org). (**C**,**D**) Expression of *HSPA1A* (**C**) and *HSPA1B* (**D**) mRNA in human ovarian cancer analyzed in (**A**). The expression levels of *HSPA1A* and *HSPA1B* were compared between samples with high *IER5* family gene expression (top 50 samples with high *IER5*, *IER5L*, or *IER2* expression) and those with low IER5 family gene expression (bottom 50 samples with low *IER5*, *IER5L*, or *IER2* expression, excluding the samples included in the top 50). (**E**,**F**) Expression of *Ier5*, *Ier5L*, and *Ier2* mRNA in mouse ovarian cancer samples derived from ovarian surface epithelium (**E**) and fallopian tube epithelium (**F**) are shown. The expression of *Ier5*, *Ier5l*, and *Ier2* were analyzed by using the data published by Zhang et al. [24]. ** *p* < 0.01, *** *p* < 0.0001.

**Figure 2 cancers-17-00610-f002:**
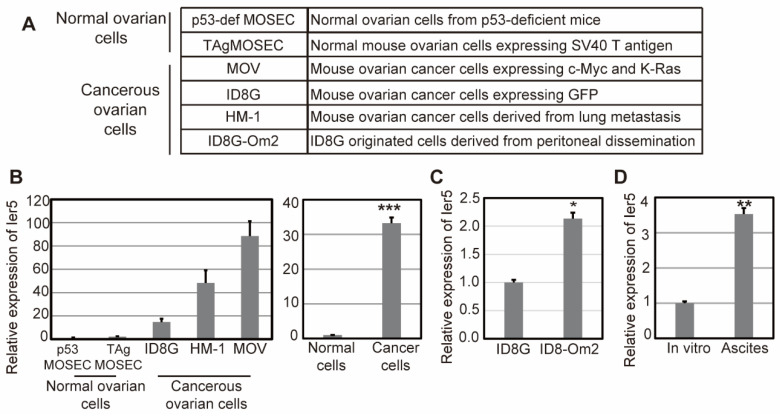
*Ier5* is highly expressed in ovarian cancer. (**A**) List of cell lines used in this study. (**B**) Relative expression of *Ier5* mRNA in *p53*-def MOSE, T-Ag-MOSE, MOV, ID8G, and HM-1 cells analyzed by RT-PCR. In the right panel, cumulative values of *Ier5* expression in normal ovarian cells (*p53*-def MOSE, T-Ag-MOSE) and cancerous ovarian cells (MOV, ID8G, and HM-1) were compared. (**C**) Relative expression of *Ier5* mRNA in ID8G and ID8-Om2 cells analyzed by RT-PCR. (**D**) Relative expression of *Ier5* mRNA in ID8G cells cultured in vitro and ID8G cells from ascites collected from mice (*n* = 4) analyzed by RT-PCR. Data are means ± SD of three replicates for representative experiments. Error bars represent mean ± SD (*n* = 3) and the ‘*p*’ values were calculated using ‘*t*’ test method. *** *p* < 0.0001, ** *p* < 0.01, and * *p* < 0.05.

**Figure 3 cancers-17-00610-f003:**
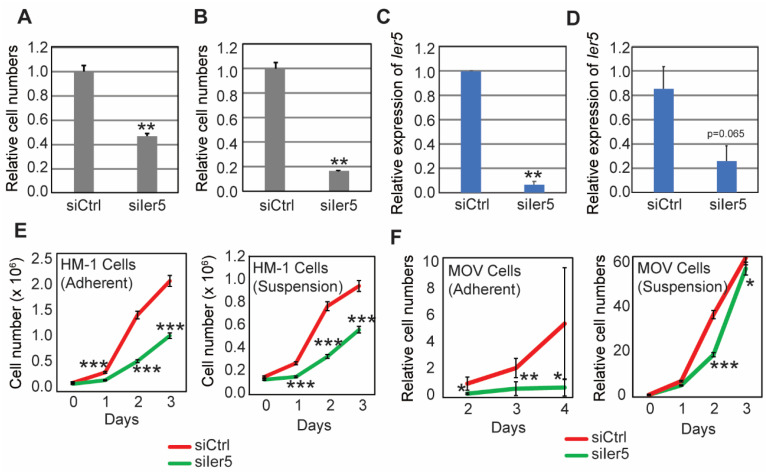
IER5 promotes ovarian cancer cell growth. (**A**,**B**) Relative numbers of HM-1 (**A**) and MOV cells (**B**) 48 h post-siRNA transfection in adherent cultures. *Ier5*-targeting or control siRNA was introduced into cells. (**C**,**D**) Relative mRNA expression of *Ier5* in HM-1 (**C**) and MOV (**D**) cells analyzed by RT-PCR. (**E**,**F**) Cell proliferation assay under adherent (left) or suspension (right) conditions of HM-1 (**C**) and MOV cells (**D**) 48 h post-siRNA transfection. Error bars represent mean ± SD (*n* = 3) and the ‘*p*’ values were calculated using ‘*t*’ test method. *** *p* < 0.0001, ** *p* < 0.01, and * *p* < 0.05.

**Figure 4 cancers-17-00610-f004:**
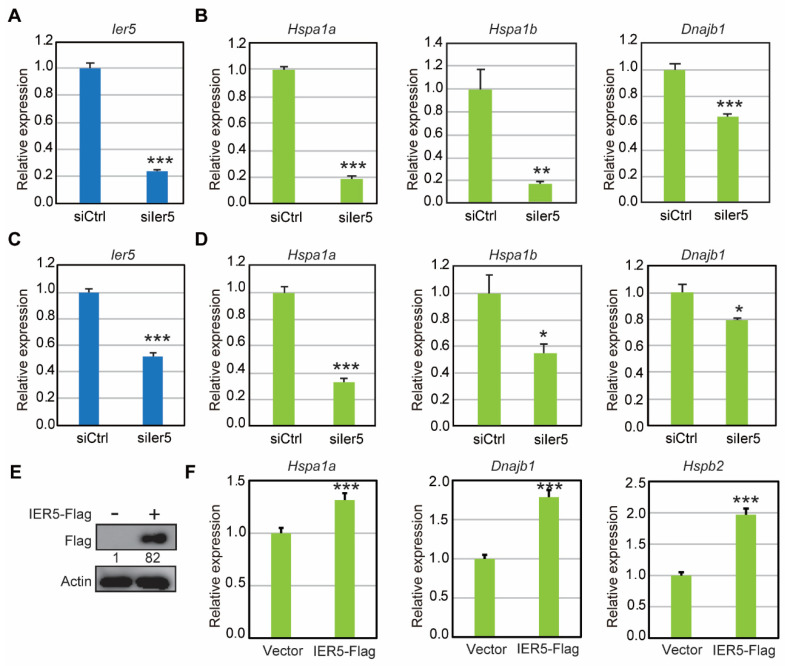
IER5 induces the transcription of HSPs. (**A**,**B**) HM-1 cells were transfected with *Ier5*-targeting or control siRNA and collected 45 h later. Relative mRNA expression of *Ier5* (**A**), *Hspa1a, Hspa1b*, and *Dnajb1* (**B**) analyzed by RT-PCR. (**C**,**D**) MOV cells were transfected with *Ier5*-targeting or control siRNA and collected 45 h later. Relative mRNA expression of *Ier5* (**C**), *Hspa1a, Hspa1b*, and *Dnajb1* (**D**) analyzed by RT-PCR. (**E**,**F**) HM-1 cells were transfected with vector or IER5-Flag plasmids and harvested 48 h later. Overexpression efficiency of IER5-Flag was analyzed by western blotting (**E**) with an anti-DDDDK antibody. Relative mRNA expression (**F**) of *Hspa1a, Dnajb1,* and *Hspb2* analyzed by RT-PCR. Error bars represent mean ± SD (*n* = 3) and the ‘*p*’ values were calculated using the ‘*t*’ test method. *** *p* < 0.0001, ** *p* < 0.01, and * *p* < 0.05. Relative densities of protein bands (except for actin) are numbered below the bands in western blotting.

**Figure 5 cancers-17-00610-f005:**
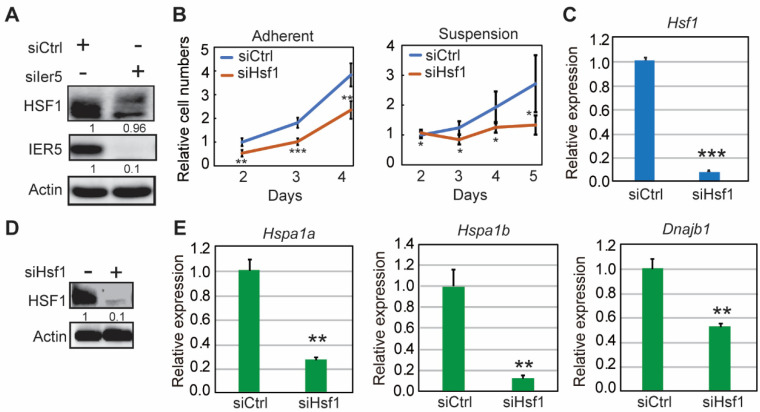
The IER5-HSF1-HS-P axis is important for ovarian cancer survival and growth. (**A**) HM-1 cells (adherent) were transfected with *Ier5*-targeting or control siRNA and collected 68 h later. Expression of HSF1 and IER5 proteins were analyzed by western blotting using anti-HSF1 and anti-IER5 antibodies. (**B**) HM-1 cell proliferation assay under adherent (left) or suspension (right) conditions for 45 h post-siRNA (siHsf1) transfection. (**C**,**D**) Relative mRNA (**C**) and protein (**D**) expression of HSF1 in HM-1 cells 45 h post-siRNA (siHsf1) transfection. (**E**) Relative mRNA expression of *Hspa1a, Hspa1b*, and *Dnajb1* in HM-1 cells 45 h post-siRNA (siHsf1) transfection. Error bars represent mean ± SD (*n* = 3) and the ‘*p*’ values were calculated using ‘*t*’ test method. *** *p* < 0.0001, ** *p* < 0.01, and * *p* < 0.05. Relative densities of protein bands (except for actin) are numbered below the bands in western blotting.

**Table 1 cancers-17-00610-t001:** List of Primers Used in the Study.

Primer	Sequence (5’-3’)
Mouse Ier5 forward	CCTTCGCTTCCAGACGATAG
Mouse Ier5 Reverse	GCGTCACCAGGTCTTTTCTC
Mouse β-Actin forward	CGGTTCCGATGCCCTGAGGCTCTT
Mouse β-Actin reverse	CGTCACACTTCATGATGGAATTGA
Mouse Hspa1b forward	CCAGTAGCCTGGGAAGACAT
Mouse Hspa1b reverse	CAGTGCCAAGACGTTTGTTT
Mouse Ier5 Forward	CGGCTCTACCCCTCTCAAGA
Mouse Ier5 Reverse	CCGAAGATGCTGATGAGGTTTG
Mouse Ier5 Probe	CCATCTCCTCGTCGGTGTCGTCCT
Mouse Hsf1 Forward	GCACACTCTGTGCCCAAGTATG
Mouse Hsf1 Reverse	AGCTGGTGACAGCATCAGAGGA
Mouse β-Actin Forward	CGCGAGCACAGCTTCTTTG
Mouse β-Actin Reverse	CATGCCGGAGCCGTTGTC
Mouse β-Actin Probe	CACACCCGCCACCAGTTCGCCATG

## Data Availability

All study data are included in the article. For further details, the corresponding author can be contacted.

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
