# Peer review of "IER5 Promotes Ovarian Cancer Cell Proliferation and Peritoneal Dissemination"

_cancers, 2025, doi:10.3390/cancers17040610_

Round 1

Reviewer 1 Report

Comments and Suggestions for Authors

The manuscript presents a study on the role of the immediate early response gene 5 (IER5) in ovarian cancer progression, proliferation, and metastasis through the regulation of HSF1 and heat shock proteins (HSPs). The topic is relevant, given the high mortality associated with ovarian cancer and the need for understanding molecular mechanisms that contribute to its aggressive nature. However, there are several areas that require substantial revision to strengthen the clarity, scientific rigor, and overall impact of the findings.

1.      In the Abstract part, the statement that IER5 functions independently of p53 in ovarian cancer is unconvincing. While it’s still debated, p53 mutants may have gain-of-function properties, so it might be best to avoid making this claim.

2.      In the Abstract part, the authors mentioned “p53-independent function of IER5 in OC”, but then “p53-IER5-HSF1 pathway”, which is contradictory and could be confusing to readers.

3.      In the Introduction part, the authors provided a brief overview of ovarian cancer but lacks specific references to previous studies on IER5 in cancers, which would strengthen the justification for this study.

4.      In the Method part, the details of siRNAs should be fully described.

5.      In Results 3.1., figure 1, the connection between IER5 alterations and patient survival should be more thoroughly discussed.

6.      In Results 3.2., lines 200-214, the rationale for selecting specific cell lines could be simplified and presented in a more straightforward way.

7.      The authors report that IER5 mRNA levels are higher in ovarian cancer cell lines than in normal ovarian cells. However, due to the variability among cell lines, it would strengthen the findings to compare IER5 mRNA levels using published TCGA data.

8.      In Figure 3, the knockdown effect of siIer5 in HM-1 and MOV cells should be tested by Western Blot.

9.      In Results 3.4., the authors report mRNA level changes in HSPs following IER5 knockdown and IER5-Flag overexpression. Adding data analysis based on TCGA datasets would enhance the robustness of these findings.

10. The manuscript claims a novel mechanism by which IER5 promotes hypo-phosphorylation of HSF1, yet the data to support this are insufficiently detailed. A thorough investigation into how IER5 and PP2A interact to influence HSF1 dephosphorylation is needed.

11. In multiple sections, claims about the functions of IER5 and HSF1 are not fully substantiated by the data presented. For example, the claim that IER5 upregulates HSP transcription should be supported by more robust gene expression data and further validated by protein-level analysis.

Reviewer 2 Report

Comments and Suggestions for Authors

The authors present an interesting study exploring the role of IER5 in ovarian cancer (OC). Showing its overexpression in cancer cells and its contribution to tumor proliferation and dissemination through the IER5-HSF1 pathway. The findings suggest IER5 and its gene family as potential diagnostic markers and therapeutic targets for OC. However, there are several critical issues that need to be addressed:

1.      Contradictory Claims: The manuscript suggests that IER5 has a p53-independent role in OC but simultaneously concludes that the "p53-IER5-HSF1 pathway" contributes to OC cell proliferation and dissemination. This is contradictory. The authors should clarify whether IER5 functions independently of or in conjunction with p53 and revise their conclusions accordingly to avoid confusion.

2.      Lack of Correlation Analysis: The cell lines used in the study do not demonstrate any clear correlation between p53 status and IER5 expression. The authors should provide additional evidence to either support or refute a link between p53 and IER5. If such a link is not established, they should adjust their interpretations and conclusions.

3.      Figure 1 Analysis: In Figure 1, the authors present data on human ovarian cancer gene alterations in IER5, IER5L, and IER2, including gene amplification and high mRNA expression. A critical question arises regarding the definition of "gene amplification" in this context. Gene amplification typically refers to a high-level copy number increase (10+ copies) and is usually associated with significantly elevated mRNA expression. However, the data in Figure 1 show that most tumors exhibit either gene amplification or high mRNA expression, but rarely both simultaneously. This discrepancy should be addressed. The authors should clarify how gene amplification is defined in their analysis and explain why amplification and high mRNA expression are not frequently co-occurring in the tumors examined.

4.      In Figure 4, the authors demonstrate that IER5 induces the transcription of HSPs. The authors performed siRNA knockdown experiments in HM-1 and MOV cells. Regarding p53 status, it appears that MOV cells are p53 null, whereas HM-1 cells have a p53 mutation. What about ID8 cells? Since this cell line is p53 wild type and exhibits significantly increased IER5 expression in ID8-Om2 cells and ascites (as shown in Figure 2), do ID8 cells show the same trend after IER5 knockdown?

5.      Furthermore, a typical western blot should be performed and presented side-by-side to confirm the knockdown efficiency of IER5 in different cell lines. This is essential for Figure 4 and should also be included in Figure 3. In Figure 5, expression of HSF1 after HSF1 knockdown also should be validated by western blot. 

Round 2

Reviewer 1 Report

Comments and Suggestions for Authors

The authors have addressed my concerns. I would like to recommend acceptance of this manuscript. 

Reviewer 2 Report

Comments and Suggestions for Authors

I appreciate the effort the authors have invested in addressing the points raised and revising the manuscript thoroughly. Overall, I am satisfied with the revisions and commend the authors for their thoughtful responses and detailed adjustments. These changes have notably enhanced the manuscript's clarity and scientific rigor.